# GnRH-Related Neurohormones in the Fruit Fly *Drosophila melanogaster*

**DOI:** 10.3390/ijms22095035

**Published:** 2021-05-10

**Authors:** David Ben-Menahem

**Affiliations:** Department of Clinical Biochemistry and Pharmacology, Faculty of Health Sciences, Ben-Gurion University of the Negev, Beer-Sheva 8410501, Israel; dbm@bgu.ac.il; Tel.: +972-8-6477485; Fax: +972-8-6479303

**Keywords:** gonadotropin-releasing hormone (GnRH), adipokinetic hormone (AKH), corazonin (CRZ), *Drosophila melanogaster* (*Drosophila*)

## Abstract

Genomic and phylogenetic analyses of various invertebrate phyla revealed the existence of genes that are evolutionarily related to the vertebrate’s decapeptide gonadotropin-releasing hormone (GnRH) and the GnRH receptor genes. Upon the characterization of these gene products, encoding peptides and putative receptors, GnRH-related peptides and their G-protein coupled receptors have been identified. These include the adipokinetic hormone (AKH) and corazonin (CRZ) in insects and their cognate receptors that pair to form bioactive signaling systems, which network with additional neurotransmitters/hormones (e.g., octopamine and ecdysone). Multiple studies in the past 30 years have identified many aspects of the biology of these peptides that are similar in size to GnRH and function as neurohormones. This review briefly describes the main activities of these two neurohormones and their receptors in the fruit fly *Drosophila melanogaster*. The similarities and differences between *Drosophila* AKH/CRZ and mammalian GnRH signaling systems are discussed. Of note, while GnRH has a key role in reproduction, AKH and CRZ show pleiotropic activities in the adult fly, primarily in metabolism and stress responses. From a protein evolution standpoint, the GnRH/AKH/CRZ family nicely demonstrates the developmental process of neuropeptide signaling systems emerging from a putative common ancestor and leading to divergent activities in distal phyla.

## 1. Introduction

The hypothalamic–pituitary–gonadal (HPG) axis is a key endocrine axis in the reproduction of vertebrates. The decapeptide gonadotropin-releasing hormone (GnRH) is secreted in a pulsatile manner from the hypothalamus and controls the biosynthesis and secretion of the gonadotropins luteinizing hormone (LH) and follicle-stimulating hormone (FSH) from the pituitary gonadotropes [1,2,3]. LH and FSH regulate steroidogenesis and gametogenesis in the gonads, and feedback loops consisting of gonadal hormones are involved in the regulation of GnRH and gonadotropin biosynthesis to tune their levels for successful reproduction [4,5,6]. It is well established that the GnRH receptor is primarily expressed in the pituitary, though expression in additional organs has been identified, and potential effects of GnRH have been documented outside the pituitary in physiological and disease states (e.g., direct activity on the prostate, mammary gland and a possible role in the placenta and gonads; for example, see [7,8,9,10,11]). The GnRH receptor is a G-protein coupled receptor, which consists of seven-transmembrane helices coupled to a G-protein and leading to the activation of an intracellular effector and generation of second messenger/s [1,2,12,13,14,15].

For many decades, the components of the HPG axis have virtually only been associated with vertebrates, though genome analyses and bioinformatics tools developed in the last 30 years enable us to identify evolutionarily and structurally related gene encoding hormones and receptors of the GnRH and the gonadotropins in several phyla of the animal kingdom (for review, see example [16,17,18]). It is generally accepted that the members of these two families, identified in different clades, probably share common roots in evolution [17,18,19,20,21,22,23]. However, their primary function has diverged in different phyla and vary, especially among invertebrate species. This review will focus on two GnRH related peptides—adipokinetic hormone (AKH) and corazonin (CRZ) [24,25,26,27,28,29]—in *Drosophila melanogaster* and emphasize their activities at the adult stage (imago). The AKH and CRZ signaling systems have been identified in invertebrates and were linked to the mammalian GnRH system based on putative similarities of the precursor gene organization and the processing of the preprohormones to the active peptides. However, the exact connection of the three ligand/receptor pairs is still not unambiguously resolved. Significant differences in the sequence and the activities of AKH and CRZ compared to GnRH (Figure 1 and see below) complicate the possibility to precisely determine the nature of the homology, and whether they are indeed “true” orthologs or paralogs, and this intriguing issue is beyond the scope of the current review. Of note, recent reviews address the evolutionary relationships with emphasis on the gene duplication rounds and loss events that led to the development of GnRH/AKH/CRZ systems and their repertoire in various metazoans [30,31]. Nevertheless, on the basis of (mainly) genomic analyses, these peptides and their receptors are generally considered as evolutionarily related and can be viewed and classified as members of the same superfamily (the GnRH family) rather than unrelated neurohormone and receptor pairs [19,21,23,30,31,32,33].

## 2. AKH Expression, Secretion and Overall Function in *Drosophila*

Like vertebrates, insects, including *Drosophila*, have numerous neuropeptide eliciting hormonal signals and complex signaling networks [34,35,36,37]. AKH is a peptidic neurohormone of 8–10 amino acid residues synthesized in the corpus cardiacum, which is part of the ring gland [24]. The ring gland is tissue connected to the brain and serves as a major endocrine organ during the larval–pupal–adult developmental phases and is viewed as the pituitary homolog in insects, including *Drosophila* [38,39,40,41]. 

In *Drosophila*, the AKH gene and peptide sequences have been reported some 30 years ago [42,43]. *Drosophila* AKH is an octapeptide, which was isolated from larvae and adults and was detected in higher levels in females [42], and although its size is about that of GnRH, the amino acid composition is different (Figure 1). About a decade later, AKH was identified as the ligand for a previously identified homolog of the mammalian GnRH receptor [44,45]. 

Unlike GnRH, the processes of the biosynthesis, the release and turnover of AKH in *Drosophila*, are largely unknown. For example, while a pulsatile secretion and half-life in the range of few minutes are hallmarks of GnRH [3,46], for the best of my knowledge, these issues have not been reported for AKH and CRZ in these flies. A very recent study in adult *Drosophila* identified multiple ion channel genes expressed in the neuroendocrine cells that synthesize and release AKH into the hemolymph when energy levels are low [47]. The mechanism of AKH secretion includes gating the K^+^_ATP_ channel and the subsequent opening of Ca^++^ and K^+^ ion channels, with a crucial role of the AMP-activated protein kinase (AMPK) [47]. The authors suggested the changes in AKH cell excitability couples nutrient sensing (e.g., low circulating sugar levels) to the exocytosis of the neurohormone to regulate metabolism, a key function of the neurohormone in adult flies ([48,49] and see below).

## 3. The AKH Receptor

A single AKH receptor was identified in *Drosophila* on the basis of various computational approaches. A putative receptor, which is structurally related to GnRH receptor, was cloned in *Drosophila* by Hauser, Sondergaard and Grimmelikhuijzan [44], and subsequently, Grimmelikhuijzan and Adams’ teams, identified AKH as its ligand [23,45]. The “de-orphanization” of the *Drosophila* GnRH-like receptor enabled them to decipher the association between AKH in insects and GnRH in mammals [19,20,21]. The AKH receptor was identified in various organs and tissues in *Drosophila*, including the fat body (the prime AKH target), the CNS and the crop (the crop is a sack-like organ; part of the gut that serves as a major storage organ for carbohydrates (and possibly for lipids) consumed by adult flies). Similar to the mechanism of action of the mammalian GnRH receptor, the *Drosophila* AKHR couples in the fat body to the Gαq/PLC pathway and Ca^2+^, an intracellular messenger of AKH [45,50]. I will summarize some major identified activities of the signaling systems of AKH and CRZ below.

## 4. Activities of the AKH/AKHR Signaling System in *Drosophila*

Initially, similar to the then already known cardio acceleration activity of AKH peptides in insect larvae, an increase in heart rate at the prepupa developmental stage was the first activity identified in *Drosophila* [42], and subsequently, diverse physiological activities have been identified up until the end of metamorphosis. Unlike in vertebrates, where the primary role of GnRH in adults is reproduction, in adult insects (imagoes), the most prominent role of AKH is related to metabolism. In adult *Drosophila*, AKH has a major role in carbohydrate and lipid metabolism, especially when high levels of energy are needed, such as in flying, locomotion and in stress conditions (e.g., starvation) [51]. The peptide induces triglyceride and glycogen break down in the fat body (the homolog of the liver and adipose tissue), increasing trehalose (a major sugar in insects) and free fatty acids and glycerol levels in the hemolymph [51]. Overall, although there is no evidence for a crosstalk at the molecular level, the energy mobilization activity of AKH is functionally antagonistic to that of the members of *Drosophila* insulin-like peptides. Hence, from a functional standpoint, based on its major function on metabolism and opposing the insulin-like system in adult flies, AKH is considered as a glucagon equivalent. In addition, the cardio stimulatory action of AKH in insects may be regarded as analogous to the direct cardiac effects of biogenic amines in mammals (e.g., adrenaline). Collectively, although evolutionarily and structurally related to GnRH, from the functional standpoint, in insects, AKH has activities similar to glucagon and adrenaline in mammals. Of note, glucagon has opposite metabolic effects compared to insulin but also a positive chronotropic effect on the heart [52,53]. Accordingly, it is generally viewed that in *Drosophila*, AKH has a glucagon-like activity without necessitating to the catecholamine.

A high-sugar diet results in hyperglycemia, insulin resistance, metabolic-like syndrome and enhanced tumorigenesis in *Drosophila* [54,55,56]. Song et al. discovered an intriguing crosstalk between Activin β and AKH in the adult fat body and showed that a high-sucrose diet (high-caloric) enhanced AKH action in the fat body, resulting in hyperglycemia [57]. Activin β secretion from the midgut is increased with a high-sugar diet, and the hormone activates the activin type 1 receptor, and this, in turn, enhances AKH receptor expression and increases AKH signaling in the fat body to promote hyperglycemia [57]. These results were consistent with a previous study by Musselman et al. that showed an increase in AKH receptor expression and enhanced AKH sensitivity in the fat body in response to a high-sugar diet and demonstrated an insulin-independent mechanism for hyperglycemia related to the diet [58]. This further supports the functional homology of *Drosophila* AKH to glucagon in mammals.

The exchange of information between the brain and the gut is important for regulating feeding, and this communication involves numerous neurotransmitters and neurohormones. A recent study identified crucial mediators of brain–gut and gut–brain communication as related to feeding and found that AKH, serotonin and octopamine modulate the crop muscle contractions in adult *Drosophila* [59]. Using electrophysiological recordings of crop muscle activity in response to exogenous stimulation of the organ with various substances, the authors reported that while serotonin increased both the amplitude and frequency of crop contraction rate, AKH mainly enhanced the frequency. In contrast, octopamine had an opposite effect, and the neurotransmitter silenced the crop motility [59]. In addition, while the accepted role of AKH is to mobilize carbohydrates from stored glycogen in the fat body, this study provided support to a previously raised possibility [51] that AKH also acts on the crop in order to increase the sugar levels in the hemolymph and provides usable energy when needed by means of pushing stored carbohydrates into the midgut for digestion [59].

Communication between tissues that involves AKH to control circulating lipid levels and energy homeostasis was also demonstrated between skeletal muscles and the intestine and fat body [60]. This study demonstrated a hormonal circuit involving the myokine Upd2 and AKH secreted from the corpora cardiacum that, depending on the muscle activity, can coordinate energy demands as related to lipid synthesis and storage. The authors suggested that this muscle–corpora cardiacum–fat–body/intestine communication enables the muscles to regulate fat turnover in response to the circadian feeding and activity changes as a mechanism to maintain the fly’s lipid hemeostasis [60].

Recently, Scopelliti and colleagues demonstrated a complex intestinal/neuronal/adipose-tissue communication network in the adult fly, in which AKH/AKHR signaling is a key switch, connecting nutrient sensing to the regulation of metabolic homeostasis [61]. This study suggested that in response to dietary sugars sensed and absorbed by the gut enterocytes (enteroendocrine cells), these cells secrete Bursiconα to circulation, leading to the activation of its receptor (dLGR2) in neurons that contact AKH neurons to regulate AKH production in the CNS [61]. The Bursiconα/dLGR2 activity results in a reduced AKH secretion from the corpora cardiacum, leading to a decrease in AKHR receptor activity in the adipose tissue in the fat body, low intracellular Ca^++^ levels and energy storage [61] (similar to the activity of the mammalian GnRH receptor, Ca^++^ is the major second messenger of the AKHR). Bursioconα is part of the Bursicon heterodimer, a key tanning hormone in insects, and its cognate receptor (dLGR2, also known as rickets, a leucine-reach repeats GPCR) are evolutionarily and structurally related to the glycoprotein hormones and their cognate receptors in mammals [16,17,18,22,62,63]. Curiously, if one tries to project the organization of the neuronal-endocrine model proposed by Scopelliti and colleagues to the messengers along the HPG in mammals, while GnRH is upstream to the gonadotropins to control gonadal function, the AKH/AKHR system is downstream to Bursiconα/dLGR2 to regulate energy homeostasis in *Drosophila*. In addition, while GnRH activity positively regulates gonadotropin biosynthesis and secretion, Scopellity and colleagues [61] proposed that circulating Bursiconα negatively regulates AKH secretion. Thus, in addition to the differences in the main physiological function, further deviation in mammals and insects exists between GnRH/gonadotropin and Bursicon/dLGR2, despite the structural and evolutionary connections between the components of these systems. 

Energy metabolism is essential for optimal reproduction, and activity of AKH in the ovaries was also reported in insects (e.g., in mosquitos [64,65]). However, oogenesis in AKH deficient *Drosophila* appeared uncompromised as judged by the similar egg laying ability of AKH and AKH receptor mutants compared to control flies, and so far, no direct AKH effects on fertility have been reported in the fruit fly [66,67,68]. Lebreton and colleagues further demonstrated the relationship of metabolism to reproduction and reported that, depending on the nutritional conditions, AKH signaling is involved in the sexual behavior of *Drosophila* [69]. In flies lacking the AKH receptor, mating behavior of males, but not of females, was reduced, especially when starved. In addition, in relation to the nutritional status, the production of certain pheromones was changed in female, but not male flies [69]. One cannot rule out the possibility of identifying a direct role of AKH on fecundity and fertility in *Drosophila* or an activity in the gonads through a secreted mediator in an analogy to the HPG axis in mammals, but thus far, these were not clearly identified. Of note, a prominent effect of the AKH/AKH-receptor on reproduction was reported in several insect species, probably by decreasing energy mobilization and vitellogenin production to support egg production (for example, in female locusts of the species *Locusta migratoria*) [70]. In consistence with this, injecting AKH into adult female crickets (*Gryllus bimaculatus*) reduced egg production (for example, in the cricket *Gryllus bimaculatus*) [71]. Furthermore, in the nematode *C. elegans*, silencing the gene encoding the neurohormone or its receptor resulted in a delay in egg-laying (this can be compared to puberty in mammals) which implies a conservation of a GnRH/AKH role for normal sexual maturation, at least in the nematode [72].

The fruit fly has been extensively used for classical developmental studies primarily because of the powerful available genetic tools in this model organism. In addition to the major metabolic effects of energy mobilization in high-energy demanding activities as discussed above, many physiological activities have been reported in adult *Drosophila*. For example, Jourjine and colleagues showed that in a small population of neurons, consisting of four neurons, AKH elicits signals of nutrient deprivation in hunger state by means of elevating intracellular Ca^++^ levels [73]. This study showed that the activities of the AKH receptor together with the cation channel Nanchung (Nan, a member of the TRPV) that senses extracellular osmolality oppositely control the fly’s consumption of sugar and water as a mechanism that regulates the drives to eat and drink [73]. 

The connection between the well-known hyperactivity of starved flies, as part of their foraging behavior, to AKH was examined in several studies. Flies devoid of AKH producing neurons or the AKH receptor did not display the locomotor hyperactivity in response to starvation, suggesting a role for the neurohormone in the hyperactive behavior associated with food deprivation [51]. Similarly, in starved adult *Drosophila* lacking AKH producing cells, trehalose levels were decreased, and female as well as male flies became hypoactive and survived longer compared to non-AKH-ablated flies under starvation [74]. Further evidence for the connection between AKH and locomotor activity was seen when knocking down AKH receptors in a small group of octopaminergic neurons (the biogenic amine octopamine is an invertebrate homolog of norepinephrine) in the fly brain, but not in the fat body, abolished the starvation-induced hyperactivity [75]. Both AKH and octopamine are crucial for the starvation-induced hyperactivity, and this study pointed to neuro-modulatory and behavioral roles of AKH in food searching when food is deficient. Subsequently, the same group showed that flies on a high-fat-diet (but not on a high-sucrose diet) became more sensitive to starvation that is manifested in an increase in the levels of the AKH receptor in these octopaminergic neurons in the fly brain and enhance their excitability [76]. This results in an hyperlocomotion behavior when starved compared to flies that were on a regular diet [76]. It was also found that a combined AKH action in the brain and fat body (in association with the octopaminergic system) fine-tunes day and night locomotion of flies with unrestricted access to food [77]. In addition, an enhanced AKH action resulted in a prolonged lifespan of water-sensing mutant female flies [78]. A recent study also reported a longer median lifespan of female AKH mutant *Drosophila* compared to control flies when fed on a high fat diet or normal diet [79]. All in all, only subtle physiological effects (including no clear effect of fecundity) and changes in the behavior and lipid status were observed during the aging of the AKH mutant [66]. Taken together, it is apparent that the despite the common evolutionary roots of AKH and GnRH, the major functions of these neurohomones and the hierarchy with additional hormones differ in vertebrates and insects.

## 5. Corazonin Activities in the Fly

Corazonin (or corazon; CRZ) is related to AKH and is an additional GnRH-like neuropeptide present in crustaceans and insects. In insects, this conserved neuropeptide is produced in neurosecretory cells and was first identified by Jan Veenstra as a heartbeat accelerator in the American cockroach, and subsequently, it was found that it has diverse physiological functions, generally related to various stress conditions. The *Drosophila* corazonin precursor gene was isolated from a genomic library and encoded a predicted signal peptide, the 11 amino acid corazonin sequence and an additional sequence of a putative peptide called corazonin-precursor-related peptide (CPRP) [25]. The structure of the corazonin precursor gene is very similar to the AKH gene, suggesting common evolutionary roots for the two neuropeptides [25,30].

Ablation of CRZ expressing neurons in the adult brain of *Drosophila* increased survival under various stress conditions including starvation, oxidative and osmotic stresses compared to genetic background control flies [80]. In addition, and possibly in relation to the stress sensitivity, manipulations of CRZ neurons affected the locomotor activity of the flies, and interestingly, there were male and female dimorphic differences [80]. The levels of the corazonin transcript, triglyceride and dopamine were also affected in CRZ deficient flies under certain stress conditions and also showed sexual dimorphism [80]. Collectively, the authors [80] suggested that (a) stress reduces CRZ signaling, in consistence with previous microarrays studies [81]; (b) CRZ neuronal function differs in males and females (which was previously observed for dopamine [82]) and (c) response to stress involves both corazonin and dopamine signaling. 

Kubark and his colleagues further showed the role of CRZ signaling in stress [83]. The authors found that CRZ receptors (CrzR) are expressed in the salivary gland and fat body of adult *Drosophila* [83]. Knocking-down CrzR expression in these organs resulted in an increased resistance to starvation and oxidative stress, an elevation in stored carbohydrate levels in starved (but not in fed) flies as well as a reduction in feeding and also had an influence on AKH and insulin-like peptide expression in the brain [83]. Based on these observations, it was suggested that peripheral CRZ affects carbohydrate metabolism, reduces resistance to various stress conditions (e.g., starvation, desiccation and oxidative stress), increases food intake and signals back to the brain to maintain metabolic homeostasis [83]. In the brain, it was shown that the fructose receptor Gr43a, which is important in nutrient sensing in the brain, is expressed in some of the CRZ neurons [84]. The authors suggested that CRZ is the likely neurotransmitter of Gr43a expressing neurons in the brain, providing the role of the neuropeptide in feeding [84]. A very recent study mapped the connectivity of CRZ and prothoracicotropic (PTTH) neurons and their projections to the prothoracic gland (PG) cells of male and female flies and also found that octopamine neurons function upstream to CRZ neurons [85]. This study showed that the CrzR activity negatively controls larval growth at a discrete advanced stage without affecting puparation by means of regulating ecdysteroid (ecdysone) production when PTTH levels are low [85]. The study deciphered an octopamine-CRZ-PTTH-ecdysone network that targets the PG and revealed a complex neuronal axis network, in which CRZ/CrzR is pivotal, that probably couples feeding to growth and maturation at the late larval stages prior to puparation [85].

It was also observed that CRZ neurons are involved in the behavioral response to ethanol in male flies. A reduced sensitivity to ethanol sedation was observed when the expression of the neuropeptide was reduced in a small population of neurons in the fly brain [86]. In CRZ neurons, the core autophagy gene, *Atg16*, has a role in the biosynthesis of the neuropeptide, regulating the ethanol intoxication behavior [87]. In addition, flies lacking CRZ neurons or the neuropeptide receptor had a delayed recovery following ethanol-induced sedation, and this extended hangover-like behavior was due to an altered alcohol metabolism [88]. Collectively, the CRZ/CrzR system apparently regulates both ethanol sensitivity and metabolism. A link between CRZ-expressing neurons and pleasurable fillings in *Drosophila* was further shown by Zer-Krispil and colleagues [89]. The authors exploited the previously identified role of CRZ in ejaculation and used several established reward parameters/assays to show that CRZ/CrzR signaling is a crucial part of the mating rewarding value in male flies. Interestingly, in the repeated activation of CRZ/CrzR signaling, the resulting ejaculation and rewarding aspects were associated with reduced ethanol consumption in the stimulated flies [89]. The authors suggested that the reduced motivation to consume the alcohol presumably resulted from the pleasurable effects of the CRZ signaling, implying a compensation between using ethanol and neurohormone signaling [89]. 

Reproduction-related activities of the CRZ system in *Drosophila* have been also reported. Tayler and colleagues showed that silencing of CRZ neurons resulted in male infertility [90]. The authors also found that CrzR expressing neurons innervate males’ reproductive organs, and the CRZ/CrzR signaling system is a crucial component in the complex coordination of copulation duration and the transfer of sperm and seminal fluid [90]. In females, while overexpression of CRZ in the brain decreased vitellinogen (egg-yolk) and egg-laying rate, its knockdown led to opposite effects, suggesting that the neuropeptide has a negative effect on female *Drosophila* fertility [91]. In contrast, a positive correlation between CRZ expression and *Drosophila* fecundity was observed by Bergland and colleagues that suggested that female fecundity is affected by the concentrations of CRZ (positively correlated) and dopamine (negatively correlated) [92]. However, another study did not find prominent differences in the number of laid eggs along a 10 day period between CRZ receptor null mutant and wild-type control flies [88]. Together, the above studies do not provide a clear generic role of CRZ in *Drosophila* reproduction and demonstrate a possible sexual dimorphism in the role of the neuropeptide in male and female reproduction. In the silkworm *Bombyx mori*, it was shown that CRZ together with GABAnergic signaling regulate the formation of diapause eggs to unsure survival when conditions are changed to unfavorable [93]. Collectively, these and additional studies show that CRZ signaling is involved in many aspects of *Drosophila* physiology, mainly metabolism, when exposed to various stresses, but also as part of a reward mechanism associated with ethanol consumption and male reproductive activities.

## 6. Summary Notes

Despite the common roots of GnRH family members in different clades, the amino acid composition and the identified activities that evolved in various phyla diverged along the evolution. While GnRH is a key hormone regulating reproduction in mammals, in insects, AKH has mainly glucagon-like activities, and CRZ is involved in stress and metabolism, but also in fecundity feeding and rewarding activities (Figure 2). One common functional issue related to the GnRH–like peptides is that they appear to work in coordination with additional neurohormones/hormones and establish elaborated signaling networks that harmonize the activity of different organs.

## Figures and Tables

**Figure 1 ijms-22-05035-f001:**
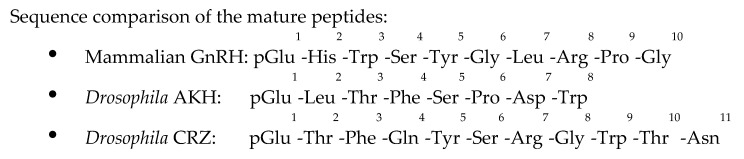
Sequence comparison of the mature GnRH/AKH/CRZ peptides. GnRH consisted of 10 amino acid residues, whereas AKH and CRZ peptides consisted of 8 and 11 amino acid residues. The amino acid residues are marked in their standard 3 letters code, and pGlu denotes pyroglutamic acid (a cyclical amino acid found in many secreted peptides). The position of the amino acid residue in the neurohormone is superscript numbered. Please note that proline is present at position 9 of GnRH and position 6 of AKH but not in CRZ, predicating differences in the secondary structure and folding of the three peptides.

**Figure 2 ijms-22-05035-f002:**
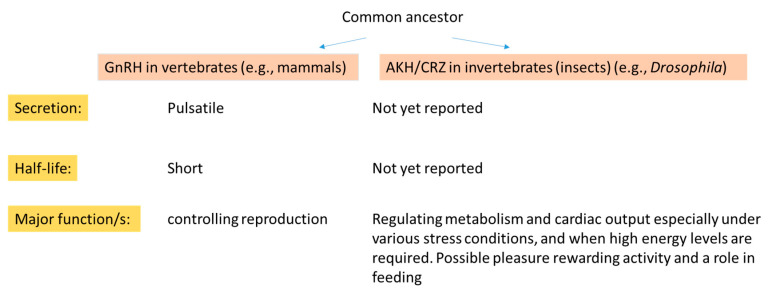
A comparison of some major aspects of GnRH/AKH/CRZ biology in mammals and insects.

## Data Availability

Not applicable.

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
