# Peer review of "GnRH-Related Neurohormones in the Fruit Fly Drosophila melanogaster"

_ijms, 2021, doi:10.3390/ijms22095035_

Round 1

Reviewer 1 Report

The review submitted by Dr. D. Ben-Menahem concerns the physiological roles of drosophila Adipokinetic hormone (AKH) and Corazonin (CRZ) and their phylogenetic relationships with vertebrate GnRH.

General comment
This review about AKH and CRZ is very interesting but their phylogenetic relationships with vertebrate GnRH is not very convincing.
I think the author should mention that these drosophila peptides were originally detected because of their putative similarity with GnRH but his review about these two peptides is interesting by itself, without sticking too much to this hypothetical phylogenetic kinship.

Specific comments

1/ As mentioned above, I am not very convinced by the phylogenetic relationships of AKH and CRZ with GnRH.

The pyroGlu end is very common in secreted peptides and, by the way, does derive from glutamic acid (as indicated in the caption of figure 1), but from glutamine (Q) present in the three peptide sequences.   Moreover, the Pro at position 6 of AKH must induce a break in the folding compared to GnRH.

2/ For a more precise phylogenetic study, the precursors of the three hormones (GnRH, AKH, and CRZ) should be taken into consideration as well as their proteolysis to release them.

3/ There are two full duplications of the genome during the evolution between insects and vertebrates. Therefore GnRH might descend, according to this view, from one of the four copies of AKH or CRZ. Therefore, wouldn’t it be interesting to look for other (maybe closer) parents of AKH and CRZ in the vertebrate genomes?

Author Response

In his general comment, the reviewer raised the issue of the phylogenetic relationships of AKH and CRZ to GnRH. I appreciate very much that the reviewer pointed this, and I absolutely agree with this questionable relationships. This issue and the specific points of the reviewer are now further emphasized at the bottom of P. 3 (L.26-31) and top of P. 4 (P.4, L.1-9). This section is now extensively modified according to the reviewer’s suggestions. The original basis for the identification of AKH/CRZ (P.3, L 28-29), that was pointed by the reviewer in his general comment, is now mentioned (P.3, L 28-29).

Specific comments:

  • The questionable phylogenetic relationships are now addressed (P.3, L.29-30). In addition, in the legend of Figure 1 text was changed to clarify the outcome of the presence of pGlu (P.13, L.25, L.27-28).
  • The GnRH/AKH/CRZ phylogenetic relationships are reviewed extensively elsewhere, and some of this major publication are cited. As above, I further refer to this point, under the limitation of the focus of the current review (P.3-4), and added an additional reference (Ref. #31).
  • I refer to the scenario described by the reviewer (bottom P.3 to top P. 4). The suggestion to look for AKH and CRZ potentially closer homologs is excellent, and should be studied in the future.

Reviewer 2 Report

Ben-Menahem described the summarized contents from the current substantial comprehensive studies concerning AKH and Crz. As the context is weak because only a few explanations for the relationship between GnRH and AKH/Crz can be seen in the manuscript. However, this manuscript is very important for this field because there are little description dealing with AKH in these couple of years as a review.

This reviewer listed the minor points which should be addressed to brush up this manuscript.

  1. Abstract

L6: which network with additional neurotransmitter/hormones: I could not understand. Please explain more.

L12: pleiotropic??

  1. Introduction

2-1

L4: the gonadotropins, luteinizing..: insert a comma.

L14: the activation of an intracellular effector…: insert a “an”.

Figure 1 caption: L2: whereas Drosophila AKH and CRZ..: insert the italicized “Drosophila”.

2-2

L8: “Drosophila” should be italicized.

L13: metabolism of AKH: “turnover of AKH” might be better.

2-3

L5: “Drosophila” should be italicized.

L6: the CNS and the crop….: AKH receptor is generally recognized to be expressed ONLY in fat body. Even if there are several exceptions, the fat body is accepted as the target of AKH in Drosophila. Author should carefully write this sentence.

2-4

L12:  “antagonistic to the members…: There are no evidence of antagonistic effects of Insulin on AKH function at a molecular level. So, author should re-arrange this sentence carefully.

L12: “Drosophila” should be italicized.

L18: the sentence beginning with “Since glucagon also have…”: The logic of this sentence seems to be strange. Author should explain this sentence carefully by dividing in several different sentences.

L19: positive chronotropic effects on the heart in mammals: references are required.

L45: Crop is the site to store the digested or ingested food. So the phrase “crop to push stored carbohydrates into the midgut” sounds strange. Author should re-arrange this sentence.

L50: “corpora cardiaca”??  Author used “corpus cardiacum” earlier. Then please use an identical term.

L60: (REF Marcos) <- what is this??

L73: the sentence beginning with “Thus, in addition to..”: Very difficult to understand. Please re-phrase this sentence.

L92: please remove “(”

L92: please remove “migratorioides”.

L94: Please remove “(for example in the cricket Gryllus bimaculatus,”.

L104: “the neurohormone”: “AKH” might be better.

L108: Please remove a sentence beginning with “The interplay between …” because of the duplicated sentences in the explanations.

L123: please remove “(HFD)” because of a single usage.

L130: resulted in an increase..: “increase” should be changed into “extension” or “prolonged life span”.

L131: “higher”: “longer” might be better.

2-5

L3: “neurosecretory cells”??

L5: “Drosophila” should be italicized.

L10: “and references therein”: please remove this phrase.

L12: including starvation, oxidative and osmotic stresses…: remove a comma after “including”.

L24: “Drosophila” should be italicized.

L25: “of the fly” should be removed.

L26: “animals” should be replaced by “flies”.

L33: the phrase around L33 should be changed as follows: “CRZ is likely the neurotransmitter of Gr43a expressing neurons in the brain..

L39: “edysone steroid” -> “ecdysteroid (ecdysone)”

L51: “Collectively, CRZ/CrzR system is apparently regulating both ethanol ….” mght be fine.

L70: “Drosophila” should be italicized.

L71: “CRZ” should be used instead of “corazonin” here.

L75: “Drosophila” should be italicized.

L77: “Bombyx mori” should be italicized.

L81: “various stresses, but also as a part of….”: “stresses”, and insert “a”.

2-6

L4: please remove “too”.

Author Response

I would like to thank the reviewer for the extensive clarifying modifications he recommended. All the points raised by the reviewer are now corrected.

More specifically:

The abstract: additional neurotransmitters and hormones are mentioned (octopamine/ecdysone) and the typo was corrected (pleiotropic).

In Figure 1 Drosophila is italicized

Introduction: The two specific points are now corrected as the reviewer asked.

Other parts of the review: All the points are addressed as requested.

Among the major issues raised and now clarified: the sentences on targeting the fat body (P.5, L. 12), the interplay with insulin like peptides (P.5, L.31- P6., L.1-2), the homology to glucagon (P.6, L. 6-11), the crop (P.7, L. 4-6 ), the sentence mentioning Bursicon/dLGR2 (P.8., L. 3-6),  CRZ and Gr43a (P.11, L.5-7) and the sentence on CRZ and ethanol sensitivity and metabolism (P.11, L.24-25) are now corrected as asked. In addition, two references are now included on the effects of glucagon on the heart (Ref. #52 and #53). The redundant sentence in the section describing the activity of  AKH and Nanchung as related to water and sugar consumption was removed (P.9, L. 2-7).

Round 2

Reviewer 1 Report

The author has considered my remarks, and I am very satisfied with his amendments to the paper.